# GRASP: Hardening Serverless Applications through Graph Reachability Analysis of Security Policies

## Abstract

Serverless computing is supplanting past versions of cloud computing as the easiest way to rapidly prototype and deploy applications. However, the reentrant and ephemeral nature of serverless functions only exacerbates the challenge of correctly specifying security policies. Unfortunately, with role-based access control solutions like Amazon Identity and Access Management (IAM) already suffering from pervasive misconfiguration problems, the likelihood of policy failures in serverless applications is high.

In this work, we introduce GRASP, a graph-based analysis framework for modeling serverless access control policies as queryable reachability graphs. GRASP generates reusable models that represent the principals of a serverless application and the interactions between those principals. We implement GRASP for Amazon IAM in Prolog, then deploy it on a corpus of 731 open source Amazon Lambda applications. We find that serverless policies tend to be short and highly permissive, e.g., 92% of surveyed policies are comprised of just 10 statements and 30% exhibit full reachability between all application functions and resources. We then use GRASP to identify potential attack vectors permitted by these policies, including hundreds of sensitive access channels, a dozen publicly-exposed resources, and four channels that may permit an attacker to exfiltrate an application's private resources through one of its public resources. These findings demonstrate GRASP's utility as a means of identifying opportunities for hardening application policies and highlighting potential exfiltration channels.

## ACM Reference Format:

Anonymous Author(s). 2023. GRASP: Hardening Serverless Applications through Graph Reachability Analysis of Security Policies. In *Proceedings of ACM The Web Conference 2024 (TheWebConf '24)*. ACM, New York, NY, USA, 11 pages. https://doi.org/XXXXXXX.XXXXXXX

## 1 Introduction

Serverless computing has revolutionized how cloud-based applications are developed and deployed. The serverless computing paradigm enables developers to focus on writing code that is uploaded to cloud providers as well-defined executable units called *functions*. The complexities of scaling functions and hardware to meet demand, load balancing traffic across functions, and system security updates are all handled transparently by the cloud provider.

However, serverless faces unique challenges as compared to traditional cloud computing. In comparison to monolithic applications, serverless lays bare the control flow of web applications creating the opportunity to perform finer-grained authorization of activities. At the same time, the serverless paradigm introduces many more security principals in comparison to a monolithic web application, increasing the difficulty of correctly specifying a least-privilege security policy. Consider Amazon's Identity and Access Management (IAM) [11], which can broadly be seen as a role-based access control model for the Amazon family of web services. IAM policies are already notoriously difficult to maintain, with a 2021 threat report finding that the majority of surveyed clients (over 63%) granted excessive permissions in their IAM policies [43]. In fact, IAM misconfiguration also played a role in the catastrophic SolarWinds breach [23]. Worse, policy misconfiguration is also extraordinarily costly to cloud customers, with a 2020 report estimating that misconfigurations have cost companies over 5 trillion USD [30]. Given these existing difficulties, how can we expect developers to make proper use of IAM in serverless environments, where policy specification is even more complex?

In this work, we present GRASP, a system for analyzing serverless security policies for potential misconfigurations. GRASP takes as input an application specification and security policy, both of which are already defined in serverless metadata (e.g., `serverless.yaml`). It then produces a reachability graph that describes the permitted interactions between application components. This graph can be queried to identify publicly-accessible resources as well as channels (paths) through which an attacker may be able to reach private resources. Using the reachability graph, developers can use queries to identify potential policy errors, e.g., *Function X should not be able to invoke Function Y, Resource 1 should not be accessible to Function Z*. After identifying such errors, the policy can then be manually updated with minimal effort – usually by updating just a handful of statements. Further, these graphs can also be used to identify functions that should be regularly audited due to high amounts of necessary privilege, or to understand how changes to the policy would impact the application's security posture. We implement GRASP for AWS Lambda, Amazon's serverless framework that accounts for roughly 75% of the serverless market [24]; further, because GRASP is based on the open `serverless.yaml` standard, it can be easily extended to support additional frameworks such as Azure, Google Cloud, and many others [47].

GRASP complements the capabilities of existing IAM policy analysis and verification tools, many of which have been developed by Amazon itself. Their ZELKOVA tool uses an SMT solver to query IAM policies, such as enumerating the subjects with permission to access a given resource [14]. Their publicly available IAM ACCESS ANALYZER provides a simplified abstraction for Zelkova, allowing developers to specifically query the subjects that have access to a given resource [12, 13]. Also built on a single class of ZELKOVA

queries, their BLOCK PUBLIC ACCESS tool detects when an S3 datastore is accidentally made publicly accessible [18]. While GRASP can also answer these resource accessibility queries, its distinguishing feature is its ability to reason about emergent security issues in serverless attacks. Given attackers' ability to compromise functions [37, 42], perform event injection [20, 36], and identify subsequent targets with desirable IAM privileges [41], it is insufficient to enumerate the *subjects* that can access an *object*; one must also consider the subjects that can *indirectly* access objects through an intermediary subject. While prior work implicitly assumes application integrity, GRASP can identify and explain these relationships through graph reachability models.

To evaluate GRASP, we collected a dataset of 731 AWS Lambda applications *enabling us to perform the first empirical study of serverless security polices*. We first confirmed that the trend to specify over-privileged IAM policies appears to have continued in serverless – 90% of applications only specify a global policy (i.e., no fine-grained permissions), 92% of policies contain 10 or fewer statements, and 30% of policies permit full connectivity between all application functions and resources. We then leveraged GRASP's various query capabilities to identify more nuanced threats. For example, we identified 14 applications with publicly-accessible resources, 227 applications with a potential attack vector for sensitive data access, and 4 applications with attack vector through which a private resource may be indirectly exfiltrated through a public resource within the same application. We supplement our analysis with four case studies of popular open source serverless applications where a manual review of the application, combined with GRASP's analysis, unearthed opportunities for dramatic privilege reduction. In summary, this work makes the following contributions:

- *We define a logic-based reachability graph model for reasoning about serverless applications security policies.* We define platform-agnostic primitives for functions, resources, and permissions, as well as relationships that capture the flow of data and execution. Our analysis engine identifies common misconfigurations (e.g., public resources) and potential attack paths to private resources. GRASP will be open-sourced upon publication.

- *Open Source Serverless Dataset.* We collect a dataset of 1,649 open source serverless application repositories, including 731 applications with valid IAM security policies. To facilitate further work in the space, our data and collection scripts will be open-sourced upon publication.

- *First empirical study of serverless security policies.* We leverage GRASP to provide the first glimpse into the start of serverless application security within open source software. Our analysis uncovers widespread use of overprivileged global policies and high levels of connectivity between application components. Through several case studies, we verify cases of overprivilege in popular web products, including an application hosted by the US Cybersecurity & Infrastructure Security Agency.

## 2 Background and Motivation

**Global vs. Function Policies:** To restrict access to sensitive data or internal services, serverless application developers specify access control policies. The two primary methods of setting permissions are through individual function policies or "global" policies that

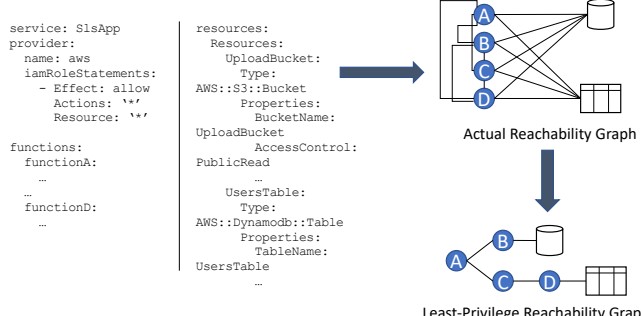

**Figure 1: Example of a Serverless Framework YAML definition with a global policy with wildcards and the resulting reachability graph.**

apply to all functions. Intuitively, a global policy is prone to over-privilege, granting individual functions access to permissions that are not strictly needed in order for them to operate. On the other hand, a policy comprised of a large number of individual function policies becomes more difficult to reason about. Both policy types can be deployed in tandem within a single application, in which case the function-level policy supersedes the global policy.

**Wildcards:** Serverless policies consist of subjects (e.g., functions), objects (e.g., resources), and permissions granted to a subject to operate on an object. Each of these components can be specified in two ways: (1) explicitly named, or (2) using wildcards. For example, FuncA is an explicitly named subject and Func* and * are wildcard subjects that match multiple subjects. Wildcard permissions are often used by developers to avoid access control issues during development when they are unsure which permissions are needed to access an object. Unfortunately, the use of wildcards can result in subjects with unintended permissions on an object.

**Managed Policies:** Managed policies are policies maintained by cloud providers and they group common permissions to assist developers who may not fully understand how to properly define a policy for their environment. While managed policies are an improvement over wildcard permissions, managed policies may still include permissions that may not be needed by the application. Further, they introduce a new problem. As these policies are not controlled by the developer themselves, any changes to the policy can have unintended consequences and reasoning about these changes is difficult. For example, in December 2021 AWS inadvertently included S3:GetObject permission to the AWSSupportServiceRolePolicy managed policy, which should only have metadata visibility [2].

**Our Approach:** In this work, we propose that IAM security policies are best analyzed through reachability graphs that describe the permissible interactions between application components. Figure 1 depicts an example of how to translate an application manifest into a reachability graph. In the left a simplified Serverless YAML file is shown that describes four functions and two resources. It can also be seen in the iamRoleStatements field that the policy is fully permissive, allowing all actions on all resources. However, even in this simplified example, it is clear that reasoning about authorizations by reading raw policy statements would become

challenging as applications and policies grow in complexity. Grasp models the policy as a reachability graph in which vertices are functions or resources while an edge denotes a particular action permitted between the two vertices, as shown in the right side of Figure 1. Of course, this reachability graph is also an information flow graph, well-known primitive for reasoning about complex interactions between principals in access control models [21, 26, 44].

## 3 Grasp

We now present Grasp, a system that *comprehensively* models application specifications and security policies as defined in `serverless.yaml` configuration files [3].

**Threat Model and Assumptions:** This work considers an adversary external to the cloud provider. This attacker's capabilities are based on the widely-established presence of bugs in both serverless function code and policy configurations – prior work has shown that policy misconfigurations enable attackers to steal sensitive information [29, 41], launch denial-of-service (or *denial-of-wallet*) attacks [1, 52], and otherwise break isolation [16]. These policy misconfigurations are exacerbated by the presence of software bugs that combine to enable event injection attacks [20, 36, 42], arbitrary code execution [50], and data exfiltration [37, 38].

We assume there are no vulnerabilities in the cloud platform's access control mechanisms that enable unauthorized users to interact with private functions and resources. By extension, we assume that the security policies enforced by the platform cannot be modified or changed by the adversary; any misconfiguration of the policy is the result of developer error. As a result, the adversary can only interact with functions and resources declared as public by the application's security policy. We make no assumption about the functions and resources themselves.

**Design Challenges:** The goal of this paper is to design a framework to model serverless applications and identify a reachability graph that explains the function-resource interactions within an application. Using this framework, developers can identify misconfigurations and critical paths in their application, allowing them to correct the policy and prioritize hardening their application against adversaries. Our approach also needs to address the gaps left by prior work where systems do not support serverless environments [27, 28] or assume function integrity [12–14, 40]. To accomplish this goal, we must overcome the following challenges:

- *Serverless applications use a multitude of methods to interact between functions and resources.* Functions can directly invoke another function or indirectly invoke another function through triggering an event. Further, functions may interact with cloud infrastructure (e.g., data storage, virtual machines, event queues, etc.) allowing functions to propagate state between each other using intermediary resources.
- *Serverless application IAM Policies are defined using multiple abstractions.* Function IAM policies can be defined globally for all functions in an application, individually on functions, or a mixture of both. Policies may be represented as a list of statements, as reusable user-defined roles, or provider-managed roles. Further, policies on a resource may also dictate what components in a serverless application can access that resource.

- *Serverless applications use proprietary cloud service APIs that are security relevant.* Each cloud provider offers proprietary services (e.g., storage, message/event queues, compute instances) and each service has unique set of APIs and permissions that enable access to invoke the different APIs (e.g., *StartInstances* and *StopInstances* actions for EC2) within that service.

**Overview:** Figure 2 depicts an overview of the architecture for Grasp. The primary design components are the (1) *Knowledge Extractor* (§3.1), (2) *Serverless Access Control Model* (§3.2), (3) *Policy Reasoner* (a Prolog engine), and (4) *Security Query* Interface (§3.3).

The *Knowledge Extractor* takes a Serverless Application Manifest and an IAM Policy as inputs and extracts facts. Functional semantics of the serverless application (e.g., functions, resources, and event triggers) are extracted from the application manifest and the permission semantics are extracted from the IAM policy. These facts create a base model for a serverless application that is fed into the *Policy Reasoner* engine.

This base model is then extended with a *Serverless Access Control Model* which is an abstraction built on top of the base model to describe higher level security concepts. These concepts are expressed as Prolog rules and capture the permissible flows of data in the serverless application, such as reading and writing resources, invoking other functions, and accessibility from the public Internet. There are broadly three categories of flows in serverless– (1) *control flows* occur when a function or public user can invoke a function, (2) *data flows* occur when a function or public user can write to a service that another function can read from and (3) *event flows* occur when a function or public user can generate some event (e.g., upload file) that automatically triggers another function to execute.

After populating the *Policy Reasoner* with facts output by the *Knowledge Extractor* and *Serverless Access Control Model*, the *Security Query* interface answers security-related queries in the form of reachability graphs. The resultant reachability graph is a set of nodes and edges where nodes represent functions and resources, edges represent flows enabled by the IAM Policy, and an attack path is a possible path from a public node to a given resource. The reachability graph describes potentially vulnerable configurations (e.g., publicly readable/writeable storage) and attack paths to a private resource (e.g., paths from a public user over the Internet to internal resources using other primitives as pivot points). This enables developers to identify errors in their application specification (e.g., a resource that should not be public) and high-risk areas in their application that should be scrutinized through careful code review (e.g., a highly privileged function with access to private resources that accepts data from untrusted users).

## 3.1 Knowledge Extractor

The Knowledge Extractor generates *application semantics* from the application manifest and *permission semantics* from the IAM policy.

*3.1.1 Modeling Application Semantics* Our serverless application model represents functions, resources, and event triggers. These primitives are translated into core facts representing the functional semantics of the serverless application.

**Functions:** Function-facts capture two pieces of information (1) *name*: the unique identifier for the function and (2) *public_invoke*: if

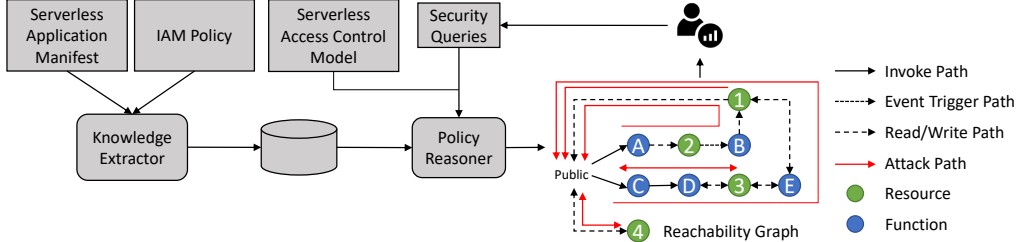

**Figure 2: Overview Architecture of GRASP.**

`true` the function can be invoked over the Internet by unauthorized users, and if `false` the function can only be invoked by authorized users using the Web GUI or CLI.

```
function(name, public_invoke).
```

**Resources:** Resource-facts capture four pieces of information: (1) *type*: the type of resource (e.g., `s3` buckets or `dynamodb` tables), (2) *name*: the unique identifier for the resource, (3) *public_read*: resource is publicly readable if `true`, and (4) *public_write*: resource is publicly writeable if `true`.

```
resource(type, name, public_read, public_write).
```

**Events:** Event-facts capture a resource that generates an event and a function that is triggered in response to the event. An event-fact may include a specific action on a resource. For example, the AWS Simple Notification Service (SNS) can trigger a function each time a new notification is written to the SNS queue. The fact for this event requires only a SNS resource identifier and the name of the triggered function. In contrast, the fact for S3 events considers an action in addition to the resource and triggered function identifiers. Example S3 actions are `s3:PutObject` and `s3:CreateBucket`.

```
event_sns_msg(sns_stream, triggered_fn).
event_s3(s3_action, bucket_name, triggered_fn).
```

*3.1.2 Modeling Access Ccontrol Semantics* Permissions to access non-public resources are captured through the different access control primitives described below.

**Permission:** Permission-facts capture three pieces of information: (1) the function the permission is granted to, (2) the permission being granted, and (3) the target resource of the statement. The permission and resource fields may contain wildcard, which matches all values.

```
permission(function, permission, target).
```

**Resource Access:** Defining if a function has read or write access to a resource not only requires knowledge of its permissions but also knowledge of the permissions that enable reading and writing for a particular resource type. This knowledge is provided as read and write permission facts. These facts are model constants specific to a cloud provider and are identified from the provider documentation. Each fact has two values: (1) the service type and (2) the permission name. The following facts are examples of read and write permissions for the AWS S3 service.

```
read_perm('s3', 's3:GetObject').
write_perm('s3', 's3:PutObject').
```

Rules defining if a function can read or write a resource use the `permission`, `read_perm`, and `write_perm` facts above and take four values: (1) a function identifier, (2) a resource type, (3) a resource identifier, and (4) a permission. For read access, the rule checks

for values, such that, there exists a `read_perm` on a given resource that enables reading and there is a `permission` defined for a given function granting the read permission on the resource target. Write access is resolved similarly.

```
can_read(Func, ResType, Resource, Perm) :-
    read_perm(ResType, Perm),
    permission(Func, Perm, Resource).
can_write(Func, ResType, Resource, Perm) :-
    write_perm(ResType, Perm),
    permission(Func, Perm, Resource).
```

## 3.2 Serverless Access Control Model

The *Serverless Access Control Model* extends application and access control semantics into *flow semantics*, allowing developers to reason transitively across functions. We define three types of flows.

**Control Flows:** These flows occur when a function or public user can pass data by invoking another function. Control flows initiated publicly are captured by the `function(···)` fact described above. Control flows for internal functions are represented by a rule with two values: an initial function `A` and a target function `B`. Using these values, the control flow rule finds a function with the given initial function name and having a permission to invoke the target function (e.g., `'lambda:InvokeFunction'` for AWS Lambda applications). Note, as a minor optimization we require that the target function is not publicly accessible; otherwise the control flow could use the public endpoint directly.

```
control_flow(A, B) :-
    function(A, _), function(B, false),
    permission(A, 'lambda:InvokeFunction', B).
```

**Data Flows:** These flows occur when a function or public user can write to a service that another function can read from. Data flows are represented by a rule with five values: an initial function `A`, a target function `B`, a resource name used to pass data, a write permission used by `A` to write to the resource, and a read permission used by `B` to read from the resource. Using these values, the rule finds an initial function that can write to the resource and a target function that can read from the resource.

```
data_flow(A, B, Resource, A_Perm, B_Perm) :-
    function(A, _), function(B, false),
    resource(Type, Resource, _, false),
    can_write(A, Type, Resource, A_Perm),
    can_read(B, Type, Resource, B_Perm).
```

**Event Flows:** Event flows occur when a function or public user can generate an event that triggers a function to execute. Event flows are represented by a rule with five values: an initial function `A`, a target function `B`, a shared resource, an action on that resource, and an event generated by the resource given the action. As an example, we

discuss how an SNS resource can be used in an event flow. For SNS resources, publishing new notifications to the resource is an action that generates a new message event. Using the values described previously, the rule finds an initial function A, a target function B and an SNS queue. Next the rule finds a permission that enables A to write to the SNS queue. Finally, the rule verifies that when a publish action occurs on the resource from A, function B is triggered. Note that the Action and the Event values are used to construct a human readable description of the flow.

```
event_flow(A, B, Resource, Action, Event) :-
    function(A, _), function(B, false),
    resource('sns', Resource, _, false),
    can_write(A, 'sns', Resource, _),
    event_sns_msg(Resource, B),
    Action = 'sns publish',
    Event = 'sns new message'.
```

### 3.3 Security Queries

Our model allows application developers to define queries to understand the security implications of their application's IAM policy. Security queries define a reachability graph of resources that can be accessed publicly or through other functions either directly or indirectly. Figure 1 shows that the queries can be viewed as operating on a graph where functions and resources are nodes, and edges encode the flows. These queries can highlight functions that need to be hardened to secure access paths to sensitive resources, and understand how modifications to an application's IAM policy change what resources can be accessed by different components.

**Publicly Exposed Components:** We begin by demonstrating how Grasp supports the "public access" query also supported by prior work [12–14, 18]. The following queries compute resources defined as public and identify common mistakes such as public S3 buckets.

```
% Publicly invokable function query.
function(Func, true).
% Publicly readable resource query.
resource(_, Resource, true, _).
% Publicly writeable resource query.
resource(_, Resource, _, true).
$ Publicly readable/writeable resource query.
resource(_, Resource, true, true).
```

The first query above is searching for all functions Func that are publicly accessible (i.e., the public_invoke parameter is true). We also make use of the Prolog anonymous variable (_) in the queries as a way to match any value. For example, it is used in the above queries except the first to match any service, because the service type is not needed to determine if the resource is publicly accessible.

**Read, Write, and Read/Write Paths:** In addition to the queries described above Grasp also supports significantly complex queries to compute indirect access paths from an application entrypoint to a given resource. A read path may allow an adversary to compromise a series of functions and exfiltrate data by embedding the data in the function response and a write path may allow an adversary to corrupt persistent storage or install a backdoor in the application. Knowing read and write paths to sensitive resources enables developers to prioritize code reviews of critical functions or remove permissions to prevent unintended paths.

Identifying these paths is achieved through the queries find_read_paths., find_write_paths, and find_rw_paths. A path from the Internet to an internal resource can start in three ways: (1) a function is public (e.g., control flow), (2) a function reads from a publicly writeable resource (e.g., data flow), or (3) a function is triggered by an event caused by a public entity (e.g., event flow). To account for these three cases, each query identifies an entrypoint for a path that determines the first function in a path and then finds a path from that function to the internal resource.

**Exfiltrate to Public Resource:** The queries above assumed that the adversary could return data in the function's HTTP response. However, that might not always be possible and in order to exfiltrate data the adversary must read data from an internal resource and then write that data to a publicly readable resource. The exfil_to_public_res query checks for paths that support this attack flow. This query first finds a read path to a private resource and then finds a write path to a public resource through the same entrypoint. Note that this query is different than the "public access" query supported by prior work – here, Grasp can identify whether a *private* resource may be written into a *public* resource as a means of exfiltration.

### 3.4 Implementation

We implemented Grasp to analyze security policies of AWS Lambda applications created with the Serverless Framework. Grasp users input their Serverless YAML definition to the Knowledge Extractor (Figure 2) to create application specific facts, then pass these facts along with the core knowledge base as input to the analysis engine. Users then perform queries to analyze their application's policy.

Grasp *comprehensively* supports Amazon Lambda configuration fields as defined in the AWS documentation. The Knowledge Extractor was implemented in Python using the checkov [6] AWS Lambda Serverless YAML parser. The generator takes a Serverless YML file as input and outputs SWI-Prolog facts. Because we are analyzing application configurations in an offline setting, extra care is taken with configuration values that are dynamically resolved, such as environment variables or command line arguments. If such values are referenced, the unresolved value is treated as a string literal throughout analysis. This is a limitation of performing offline analysis on a large dataset and is discussed further in Section ??. Once fully parsed, the serverless definition is used to build the knowledge base. Section A describes special implementation considerations in generating the facts for our experiments.

## 4 Open Source Serverless Dataset

Although serverless success stories are prominently advertised in industry [5, 7, 8], these applications are proprietary and thus not available for analysis. As we are not aware of any public dataset of serverless applications, we create the first such dataset based on the serverless applications available on GitHub. This dataset will be made publicly available upon publication.

**Data Collection:** Our collection tool was a GitHub scraper written in Python and focused on applications built using the Serverless Framework [3], a popular framework for building serverless applications for deployment on the major cloud providers. In order to identify whether a repository contained a serverless application, the scraper searched for a Serverless Framework definition file.

The GitHub scraper collected data in two phases. In the first phase, the scraper identified all projects on GitHub with at least 10 stars to filter out low quality applications. Phase 1 was performed

over a 2 day period from 9/16/2021 to 9/17/2021. During this time the scraper identified 1,292,868 repositories with at least 10 stars. It was estimated that there were 1,374,108 repositories at this time but networking limitations prevented the full collection.

In the second phase, the scraper searched the identified repositories for a Serverless Framework YAML definition (e.g., `serverless.yml`) using the GitHub API. Phase 2 was performed over a 8 day period from 9/29/2021 to 10/6/2021. Of the 1.3 million repositories, 1,649 contained a serverless application. Of the serverless application repositories, 578 (35%) of the projects defined an IAM policy. Within the repositories that contained an IAM policy, we identified 1,064 `serverless.yml` files, 731 (68.7%) parsed without error.

**Parsing Failures:** We manually reviewed the 343 (31.3%) configuration files that encountered parsing errors. 289 failures were due to the configuration files being invalid in one way or another, causing the underlying parser [6] to fail. The remaining 54 files query a live API, e.g., using `Fn::ImportValue`. Because we do not have access to these API's, we configured GRASP to throw a warning and abort in these cases. In practice, however, developers will be able to access their own API's, so this limitation is experimental rather than methodological.

**Remarks:** Our scrape of GitHub serverless applications identified three classes of IAM policies that will become relevant in the remainder of our analysis. Of the 731 applications, security policy is specified in one of the following three ways: (1) **DS1:** Applications defining only a global IAM policy for functions (658 total). (2) **DS2:** Applications only defining function-specific IAM policies (49 total). (3) **DS3:** Applications using a composition of global and function-specific policies (22 total).

Applications in **DS1** necessarily define a course-grained policy due to the fact that all functions possess the same permissions. In contrast, applications in **DS2** and **DS3** may define finer-grained permissions such that not all functions possess the same privilege set. In the following section, we explore the repercussions of coarse-grained security policies.

## 5 Characterizing Serverless Security Policies

Using our dataset of open source serverless applications, we now leverage GRASP to provide further insight into the security policies defined for AWS Lambda applications.

### 5.1 Application Complexity

We begin by describing the complexity of applications in terms of the number of functions and data resources identified in their reachability graphs. The cumulative density of functions per application is given in Figure 3a. While up to 35 functions were seen in a single serverless application, the vast majority of applications (91%) have 5 or fewer functions. Across the three data sets, 608 (92%) of **DS1**, 38 (77.6%) of **DS2**, and 17 (77%) of **DS3** applications have 5 or fewer functions defined.

The cumulative density of data resources per application is given in Figure 3b. As with function definitions, we observe a long tail in which some applications define up to 14 distinct data resources. That said, 721 (99%) applications have 5 or fewer resources and the median number of resources per application is just 1. A notable number of applications (224) define no resource at all, leaving just

485 in our dataset with at least one resource. Interestingly, many more applications in the **DS2** group feature no resource at all as compared to the **DS1** and **DS3** groups.

### 5.2 Policy Complexity

We now consider the complexity of the security policies, both in terms of the number of statements they include and the precision with which the resources are authorized.

**Statement Density:** Figure 3c describes the number of policy statements per application by policy type (Global-Only, Function-Only, Global and Function). Developers can define statements that describe a single permission on a single resource and more complex statements that describe a list of permissions on a list of resources. The latter is actually translated by IAM into a cross product of individual statements between the permissions and resource lists. For our analysis we count the individual number of statements generated as a result of these expanded statements, when applicable. As can be seen, the vast majority of policies ($n$=670, 92%) are comprised of 10 or fewer statements. In fact, policies contained 10 or fewer statements in 91.8% of **DS1** ($n$=604), and 96.0% of **DS2** ($n$=47), and 86.4% of **DS3** ($n$=19). The largest observed policy was in **DS3**, comprised of 76 statements.

**Function-Resource Connectivity:** The permissiveness, or connectivity, of a given application could be described as the number of authorized function-to-resource flows relative to the number of possible function-to-resource flows. Specifically, we calculate an application's connectivity ratio as $\frac{|\text{Function-to-Resource Flows}|}{|\text{Functions}|\cdot|\text{Resources}|}$. Figure 4 reports on the cumulative density of connectivity ratios in our dataset, considering both direct connections and indirect connections. Over 30% of applications have a connectivity ratio of 1, indicating a highly permissive security policy.

**Wildcards:** Recall that, in a given policy statement, wildcards can be used in one or both of the permission or resource fields. We distinguish between two kinds of wildcard use in our analysis: (1) a full wildcard (i.e., `*`) and (2) a partial wildcard (e.g., `s3:*`). The use of wildcards can not only lead to overly permissive policies by introducing unintended permissions, they also make reasoning about the correctness of a policy more difficult. Fortunately, GRASP is able to precisely quantify the effect of these wildcards by expanding wildcards into concrete statements applied on a subject and object.

Across all applications, 6 policy statements featured a full wildcard permission and 347 statements featured a full wildcard resource. We observed 2 statements in which a full wildcard permission was granted to a full wildcard resource. For partial wildcards, we found 160 policy statements in which they were used in the permissions field and 41 statements in which they appeared in the resources field. We observed just 1 statement that featured a partial wildcard permission to a partial wildcard resource. However, collapsing across the two wildcard types, we observed a total of 130 policy statements in which a wildcard permission was specified for a wildcard resource. This once again stresses the potential of creating overly permissive policies and enabling functions to access resources or other functions with no legitimate business purpose.

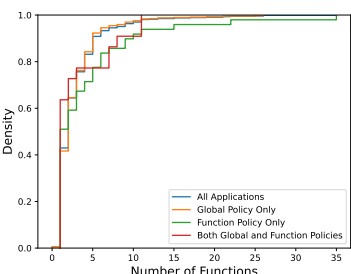 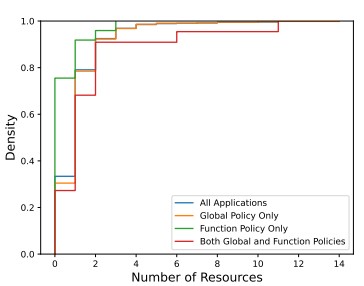 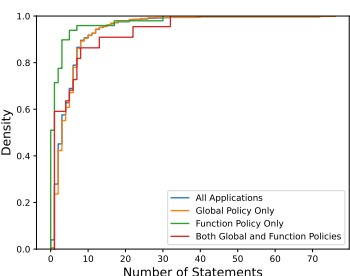

**Figure 3: Figure 3a describes cumulative density of serverless functions per application. 91% of applications define 5 or fewer functions. Figure 3b describes cumulative density of data resources per application. 99% of applications define 5 or fewer resources. Figure 3c describes cumulative density of policy statements defined per application. 92% of applications define fewer than 10 statements.**

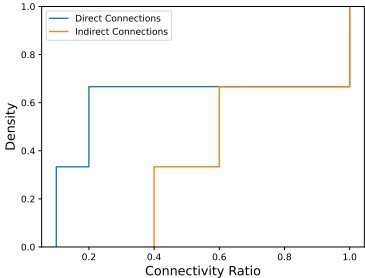

**Figure 4: Cumulative density of connectivity ratios in the analyzed Lambda applications.**

### 5.3 GRASP Queries

**Publicly Exposed Components:** Fully public application components are especially sensitive because they can be directly accessed by the attacker. Of the 485 applications in our dataset that define at least one resource, 346 contain only privately-defined resources. Of the 476 applications that define a publicly invocable function, 342 (71.8%) also define a private resource. Verifying the correctness of security policy is especially important in these cases, as permissive policies may permit indirect access to the private resource.

Data resources that are directly accessible are especially sensitive. We identify 13 applications with a read-only public resource, 1 with a public read-write resource, and 0 with a write-only resource. A publicly accessible resource is not necessarily a policy misconfiguration; for example, a readable resource could host download images that could be directly linked from other websites. However, carefully auditing public resources is vital to application security. GRASP simplifies this task by unifying distributed security policies into a single graph.

**Read, Write, Read/Write Path:** Non-public application resources also may be vulnerable if a workflow path exists in which attacker-controlled inputs cause the resource to be accessed. GRASP identified 227 applications that include at least one read path, 243 applications that contained at least one write path, and 219 applications that

contained at least one read/write path. The existence of such paths does not necessarily imply an IAM policy misconfiguration, but does suggest that the components in the path must be considered as part of the attack surface of the application. By enumerating these paths, GRASP can enable the developer to comprehensively audit permissions to determine if the discovered paths are strictly necessary for the application's business logic.

**Exfiltrate to Public Resource:** Even when attackers gain read access to a private resource, they must identify a policy-permitted method of exfiltrating the data. GRASP identified 4 applications that contained a path in which a public user could invoke a publicly accessible function to read from a private resource and then write to a publicly readable resource. While the existence of these paths do not guarantee a vulnerability, it is important that paths from private resources to public resources be carefully audited.

**Performance:** We briefly remark on the speed of GRASP offline policy analysis. Of the 731 application policies analyzed, all but one finished in under a second. The remaining application was a proof-of-concept application for storing time-based events in DynamoDB[1]. Policy analysis for this application took 3.9 seconds. Even in the most extreme cases, GRASP is able to return results for an IAM policy extremely quickly.

### 5.4 Policy Hardening Case Study

Next we will explore the difference between an overly permissive policy and a hardened policy through case studies. We selected four practical applications that have recent GitHub activity. We describe one of them in detail here, and discuss the rest in Section B.

The *Shan18/Flash* application creates an end-to-end Deep Learning platform that allows users to create, train, and deploy their own neural network models. GRASP generated the paths to important resources in *Shan18/Flash* application according to the original policy defined within it. The original policy use global permissions and reveal a number of privilege violations when manually analyzed. We describe our findings from analyzing *Shan18/Flash* below followed by a suggested hardened policy for this application that adheres to

---

[1]https://github.com/alessandrobologna/dynamodb-event-store

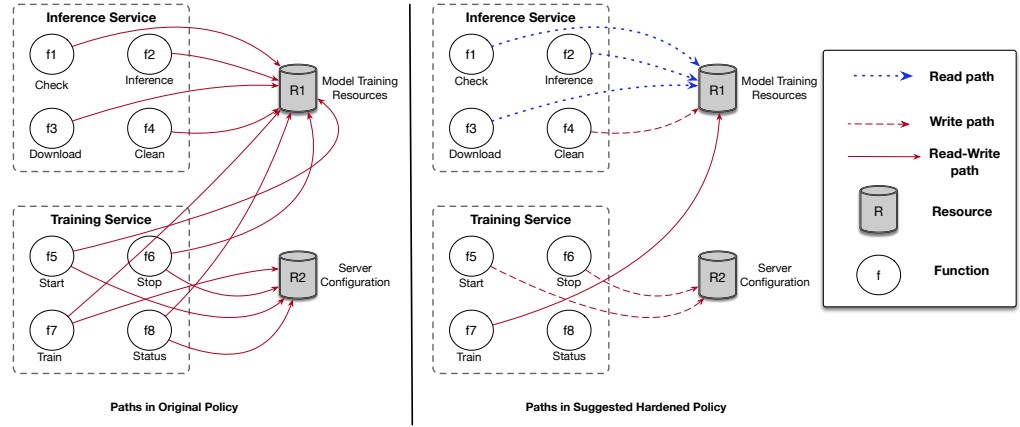

Figure 5: Grasp discovered all access paths in original policies defined in the *Shan18/Flash* application as described in the left-hand side of the figure, and helps in creating a hardened policy that only allows the intended access paths required as shown in the right-hand side.

the principle of least privilege. The hardened policy was created by manually reviewing the application source code and reviewing which functions interacted with the public, other functions, and resources. Only functions that accessed resources or other functions in code were granted permissions that enabled them do so.

**Shan18/Flash:** The Flash application implements two service workflows, namely training and inference, consisting of eight functions and two resources, as described in Figure 5. The described policy of this application allows full global access to its resources and, resulted in the access paths shown in the left side of Figure 5. After examining these paths and reviewing the application code, we found out that, only the *train* (f7) function requires full access to R1, i.e. the model training resources. Similarly, other accesses in the original policy can be trimmed down to enforce only read accesses for *check* (f1), *inference* (f2), and the *download* (f3) functions to R1; write access for *clean* (f4) to R1 and read-write access for *train* (f7) to R1; write access for *start* (f5) and *stop* (f6) functions to R2, i.e the server configuration resource. Consequently, our suggested least privilege policy following the Grasp-analysis dropped the number of paths to 3 read paths, 3 write paths, and 1 read/write path.

## 6 Related Work

Our work joins a rich literature that draws on formal methods to analyze security policies and identify misconfigurations. We have already highlighted prior analysis of IAM policies [12–14, 40]. Alternative security modules such as Tomoyo [31] and SubDomain [22] were specifically designed with the goal of enabling simpler specification of coarser-grained policies. Wang et al. demonstrate through their EASEAndroid system that policies can be automatically refined using semi-supervised learning [51]. Some notable works include verification of the security properties of SELinux policies [32, 34, 48], mandatory access control security policy analysis to identify the attack surface of applications [49], to reduce software measurements in attestation [33], and to reduce the number of subjects in auditing policies [17]. Beyond access control policy analysis, formal methods have been employed for privacy compliance

[28, 46] and social network policies [39]. Our work is most clearly distinguished from these past efforts through its explicit support for serverless frameworks and ability to reason about complex attack paths through indirect resource access.

Researchers have considered serverless flow control (e.g., Trapeze [10], Valve [25], SecLambda [35]) and other access control models (e.g., Will.IAM [45]) to deny suspicious access requests on-the-fly. Orchestration frameworks have also been enhanced [19] with security policy support for serverless applications. While not a security mechanism in itself, Obetz et al.'s method of construction call graphs of serverless applications [40] may identify security vulnerabilities; note that these graphs are based on dynamic analysis of serverless control flows, not policy-based information flow graphs like Grasp's. While not designed for serverless, Baig et al.'s Cloudflow presents methods for cloud platform information flow control through VM introspection [15], while McCune et al.'s Shamon system demonstrates how to extend reference monitor guarantees into distributed environments [17]. However, the above works do not address the risks of policy misconfiguration in today's widely-used access control models such as Amazon IAM.

## 7 Conclusion

This work presented Grasp, an automated policy analysis framework for serverless applications. Grasp defined a logic-based model for serverless security policies that captures the flow of data and execution within a serverless application. This model can then be used to produce reachability graphs that identify potential misconfigurations and paths to resources in an application. Using this framework, Grasp performed the first empirical study of open source serverless security policies. Our study discovered that serverless policies tend to be short and highly permissive, e.g., 92% of surveyed policies comprised of just 10 statements and 30% of them had full reachability among all application functions and resources. This study demonstrates Grasp's utility in identifying potential exfiltration channels and lays the path forward for further research in policy hardening.

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

## A  Special Implementation Considerations

While GRASP was designed to be independent of the underlying technologies and cloud platforms, GRASP is implemented for serverless applications running on the popular AWS Lambda platform and specified using the widely used Serverless Framework. Currently, GRASP handles the following AWS services: S3, DynamoDB, SNS, SQS, IoT, CloudWatch, logs, Kinesis, Sagemaker, SDB, AppSync, RDS, Neptune, and Lambda. These services all enable control, data, or event flows and are common services used in applications. In the future, GRASP can be extended to other platforms (e.g., Azure Functions or Google Cloud Functions) or other deployment frameworks (e.g., Terraform). To expand to these other platforms, primitives for each of the platforms will have to be added to the core knowledge base and the graph generator will have to understand and parse specifications in other formats.

**Functions and Events:** Functions are defined in the YML `functions` object. In Lambda, functions can only be invoked from the command-line or web GUI by authorized users unless there are events defined. Therefore, when building the function facts, all functions start out as initially private until an HTTP(S) event or API event is encountered in the YML. During the parsing of events, the generator also builds event facts such as `event_sns_msg(·).` and `event_s3(·).`

**Resources:** GRASP considers two types of resource definitions. First, there are explicitly listed resource in the `resources.Resources` object. Second, there are implicit resources that are used in IAM statements but not defined in the YML. To build the resource facts, a first pass is done to collect all explicitly named resources, applicable resource policies, and implicitly named resources from IAM statements. Once all resources and resource policies are collected, the generator checks if there is a policy for each resource. If no policy is found, the resource is considered private and the public cannot read or write to the resource. However, if a policy is found the public read and write permissions are extracted from the policy.

**Permissions:** Permissions for functions can be defined as global statements, individual function level statements, within a user-defined role resource applied to functions, or within platform provider managed roles applied to functions. Permission facts are built by first collecting all the IAM role statements at the global level, then the generator checks for defined roles in `resource.Resources`, and finally at the function level. If no role or permissions are defined for

a function, the function is assigned the global statements. If a role is defined for the function, the function is assigned the statements from that role. Finally, if permissions are defined at the function level, those statements are assigned to the function. Note, GRASP supports legacy statement definitions, the current statement definitions, and the Serverless IAM Roles Per Function Plugin [4] that extends the framework and enables role statements to be declared on an individual function.

We manually identified 42 read and write permissions across 13 AWS services that store or transmit data in November 2021 using the AWS documentation. Note, the list of 13 services is not complete and does not cover every service offered by AWS. This work only focused on services used by the applications in the data set. Additionally, we identified 2 permissions related to control flows and an S3 permission related to event flows. Further, we expand all platform managed roles (e.g., AWS Managed Policies) into a complete list of permission statements.

**Read and Write Permissions:** As mentioned above, we manually identified 42 read and write permissions across 13 AWS services, 2 permissions for invoking functions, and a S3 permission for deleting object that would cause an event trigger. This process needs to be completed for each service and provider supported by GRASP. Initially, we focused only on the services that were defined and used the applications in our data. We envision expansion to other services and providers through automatically identifying supported permissions as an important future work.

Once the application specific facts are generated, they can be combined with the core knowledge base and queries can be performed. The GRASP implementation generates SWI-Prolog facts. The core knowledge base follows the base design described above, but it was extended in three ways for better usability. First, additional variables are used to create human readable path descriptions to better understand an attack path and what permissions are used at every hop. Second, an overall length is reported that counts the number of functions that would need to be compromised in order to use that path in a successful attack. Third, a count of control flows, data flows, and event flows is reported for each path. The last two extensions also enable the developer to specify a how long a path should be (e.g., only show paths of length 1) and how many flows should be present in the path (e.g., do not include data flows or event flows in resulting paths). Using these tuning variables, developers can exclude paths that are outside of their risk model and less likely to happen.

## B  Additional Policy Hardening Case Studies

- **Cisagov/Crossfeed** This is an application that monitors an organization's public-facing attack surface in order to discover assets and potential security flaws and allows customers to view scan reports.
- **Arackaf/Booklist** This web application keeps track of user's book collections and enables searching, tagging, and organizing books with hierarchical subjects.
- **Connorads/Lockbot** This applications helps coordinating the use of shared resources in an organization's communication platforms (e.g., Slack).

We analyzed the above application with Grasp and generated the paths to important resources in these applications according to the original policy defined within the applications. The original policies use global permissions and reveal a number of privilege violations when manually analyzed.

**Cisagov/Crossfeed:** The policy for this application enables lambda invoke, S3 GET and PUT, and SES send permissions for all functions. This policy resulted in 1 read, write, and read/write paths. Note that Grasp identifies only the shortest path to each resource. Since the entrypoint function api has full access to all resources the only path considered by Grasp is from the Internet through api and directly to the single s3 bucket. If we remove this optimization and consider all possible paths, using different read/write permissions for each path and different intermediate functions, we find the original policy enabled 6,331 paths and our suggested policy remains at 1 read, write, and read/write path. This shows the original policy enabled many non-required flows.

**Arackaf/Booklist:** The policy for this application enables DynamoDB, S3, and Secret Manager access and resulted in 9 read paths, 9 write paths, and 9 read/write paths. Our suggested least privilege policy dropped the number of paths to 5 read paths, 5 write paths, and 5 read/write paths.

**Connorads/Lockbot:** The policy for this application enables DynamoDB access for all functions and resulted in 9 read paths, 9 write paths, and 9 read/write paths. Our suggested least privilege policy dropped the number of paths to 7 read paths, 7 write paths, and 7 read/write paths.

## C  Dataset Limitations

The dataset collected and used in the empirical evaluation has the two primary limitations. First, the dataset may not be representative of how actual developers use serverless in practice. While our goal was to create the first known dataset of serverless applications, a well-known issue with serverless research is the lack of public production quality serverless applications. While there are many success stories of serverless being used in industry, very few of these applications are made public; therefore the applications in the dataset may not represent how developers are using serverless IAM policies in production. Even with this limitation, the dataset does capture public examples of serverless applications and problems found within this dataset could potentially influence developers new to serverless development to propagate the same issues. Second, our dataset only explores the use of the Serverless Framework for AWS Lambda applications. Future work should extend to support other frameworks (e.g., Terraform [9]) and cloud providers (e.g., Google Cloud Platform and Microsoft Azure).

