# OpenReview forum: "GRASP: Hardening Serverless Applications through Graph Reachability Analysis of Security Policies"
_ACM.org/TheWebConf/2024/Conference — TheWebConf24_

### Official Review · Reviewer_r2WV · 2023-10-24

**Novelty:** 2
**Technical Quality:** 5

**Review:**

## Paper Summary

The paper presents GRASP, a Prolog-based framework for analyzing access control policies for the Lambda serverless platform. GRASP allows querying the IAM policies for potential vulnerabilities, including (i) publicly exposed resources, (ii) read/write paths, where a public function can write/read a private resource, and (iii) exfiltration to public resources, where a public function stores in a public resource data obtained from private resources.
The authors evaluate GRASP on a dataset of 731 open-source applications, discovering the widespread use of over-privileged policies, and then discuss representative case studies.

## Strengths

- GRASP allows to query a Serverless Framework YAML file for potential vulnerabilities in the access control policy definitions.
- The authors construct a dataset of open-source applications using the Serverless Framework that can be used for future research.

## Weaknesses

- The completeness of the GRASP w.r.t. AWS features and services should be clarified, discussing (and quantifying, if possible) the potential for false negatives.
- Security implications of the GRASP results should be highlighted in the case studies. How many of the potential vulnerabilities reported by GRASP correspond to actual attacks?
- It is unclear how GRASP's approach to building the knowledge base compares to previous work [13] on the generation of positive facts defining permissions.

## Comments

The paper addresses the important problem of identifying the security issues that emerge from the graph-like shape of access control policies, allowing to reason about indirect access to resources. However, the paper could be improved by clarifying the completeness of the tool, discussing potential limitations of the implementation, and showcasing how GRASP results map to concrete attacks on serverless applications.

### Completeness

- In Sec. 3.4., the authors state that GRASP comprehensively supports Lambda configuration fields. However, Appendix A specifies that only 13 AWS services are covered regarding permissions. Since the definition of the access control model (Sec. 3.2) and queries depend on the supported services, the tool answers are affected by its completeness, potentially resulting in false negatives. Reporting the supported AWS features, e.g., in a table, and adding a discussion quantifying the possibility of false negatives, would improve the paper by clarifying the scope of the tool.

- How are AWS-managed policies integrated into the knowledge base? Can such policies be automatically updated, or do they require manual intervention? More specifically, in the case of misconfigurations on the provider side (as in the example highlighted in Sec.2 [2]), can GRASP notify the user?

### Implementation Limitations

- The generation of permission and resource facts requires transforming the policy, containing a combination of Allow and Deny rules operating on potentially wildcard resources, into a set of positive Allow-only statements. How does the knowledge extractor perform this transformation?
  - Sec. 5.1 states that wildcards are expanded into concrete statements; however, the methodology is not clearly explained, especially in the case the expanded statement set contains both an Allow and a Deny rule for the same (subject, resource) pair.
  - The generation of such Allow-only facts resembles the output of AccessAnalyzer [13], which uses ZELKOVA [14] as an oracle for determining which access is allowed. How does the knowledge extractor of GRASP compare with AccessAnalyzer?

- Sec. 3.4 discusses how GRASP treats dynamic values (e.g., environment variables) as string constants, referencing a missing section for the impact of such limitation on the analysis.

### Security Implications of the over-privilege in the selected case studies

The paper presents case studies of overly permissive policies in open-source applications, proposing a hardened policy that follows the principle of least privilege for each one. The creation of this policy requires the manual analysis of the application code to characterize its permissions requirements. The difference between the original and the hardened policy shows that specific rules are not needed during normal operations. However, it is unclear if such additional rules can be exploited in the studied application to perform concrete attacks (e.g., leaking private data).

## Editorial Remarks

- 3.3 "Figure 1" => "Figure 2".
- 3.4 "Section A", 5.4 "Section B" => "Appendix A", "Appendix B", respectively.
- 3.4 "Section ???" => Add the referenced section. See the comment above.

## Update After Author Response

The authors plan to discuss the completeness of the tool w.r.t. supported AWS services, discuss the security implications of GRASP results by presenting a case study of real-world exploitability, and add an example of the difference with Zelkova. These changes address all the weaknesses highlighted in the review.

**Questions:**

- Is GRASP able to reason about policy changes in AWS-managed policies?
- How many of the potential vulnerabilities reported by GRASP can be exploited in the presented case studies?
- How does the GRASP knowledge extractor component compare to previous work, especially AccessAnalyzer [13]? In particular, how are positive permission facts generated from IAM policies containing wildcards and Allow/Deny rules without relying on a solver?

**Ethics Review Description:**

--

**Reviewer Confidence:**

3: The reviewer is confident but not certain that the evaluation is correct

**Scope:**

4: The work is relevant to the Web and to the track, and is of broad interest to the community

---

### Official Review · Reviewer_zMds · 2023-10-26

**Novelty:** 4
**Technical Quality:** 4

**Review:**

## Pros

+ An interesting, often-ignored problem space in serverless computing
+ Some interesting findings about private resource leaks

## Cons

- It remains unclear to me how significant those finds are (e.g., what consequence they could lead to and what private sources are)
- No developer feedback is obtained
- No consideration of serverless code but only policies

## Comments

Overall, I think this is an interesting paper and has a place in theWebConf.  I like the unique angle of analyzing security policies in serverless computing.  The findings are also interesting in terms of several possible privacy concerns, e.g., leaks of private resources.  At the same time, I also have some concerns:

First, the paper only considers the reachability from the function level.  However, unique code inside functions are not analyzed.  It is possible that the path is reachable, but the real code would prevent such a path (e.g., with a sanitization or a permission check, etc.) I understand that this is an upper bound, but it would be nice to show some lower bound too.

Second, the malicious consequence of the discovered results would be largely unknown.  The authors categorize them as "potential attack vectors", but it is unclear whether they are indeed exploitable.  Also, the authors say "four channels that may permit an attacker to exfiltrate an application’s private resources through one of its public resources. " It is unclear to me what private resources are and how sensitive they would be.

Third, while the paper discovered many potential security issues, it is unclear how developers are reacting to such findings.  Also, it is unclear how popular these serverless computing policies are (e.g., how many stars these serverless computing applications have on Github).  It would be nice to look at popular serverless computing applications (instead of something that no one uses).

**Questions:**

1) How popular are these serverless computing apps in your dataset (e.g., how many github stars are those vulnerable ones)?

2) Did you contact the authors and obtain their feedback?

3) What if you consider code (e.g., manually) in addition to the policy?

**Reviewer Confidence:**

2: The reviewer is willing to defend the evaluation, but it is likely that the reviewer did not understand parts of the paper

**Scope:**

4: The work is relevant to the Web and to the track, and is of broad interest to the community

---

### Official Review · Reviewer_Cjmy · 2023-11-22

**Novelty:** 6
**Technical Quality:** 3

**Review:**

## PROS/CONS:

- (PROS) The paper presents an interesting approach to formalize and analyse the access control policies of serverless applications.

- (PROS) The work is original and significant for the conference.

- (PROS) The paper is well written

- (CONS) However the paper flow would benefit IMHO of more examples to provide more concretiness to the reader. I would suggest the authors to add a real motivating example
and use it through the paper to present the different concepts and formalization.

- (CONS) The **main problem** with the paper is the weak validation of the approach that also make unclear how impactful the approach can be in the real-world.
I totally understand that the authors cannot access real industrial serverless applications and that they did their best to collect
public-available applications on github to build an initial dataset.  However, the results provided by GRASP could have been validated with some
of the development teams in charge of these projects in github. I would not expect all the development teams to willingly contribute, but some
may have done it, especially those for more popular projects. By the way, it is unclear how popular are the 731 that you could add to your dataset
(we only know that they have more than 10 stars)? You could start with those 4 apps that you have already hardened yourself, also to get confirmation
from the developers. Another interesting direction would be to perform a user-study with 10-20 developers to evaluate the impact of your approach.

## Other comments:
- You may want to add an example of one of your complex queries and add the others in the appendix
- in the Implementation section there is a broken reference to "Section ??"

**Questions:**

- validation: is there any chance to validate your results with (some of) the development teams responsible for the apps in your dataset?
- dataset: from your scraping only ~1/1000 app is a serverless one. However the scraping was done in 2019. Any difference today? More apps? How popular are they?
- unresolved value: how many times the limitation on unresolved value impacted you in your dataset?
- results: would it be possible to provide a few complete examples (maybe from one of the 4 apps you analysed in details) of a path reported from your approach
that is really a problem and one that is not?

**Ethics Review Description:**

n.a.

**Reviewer Confidence:**

3: The reviewer is confident but not certain that the evaluation is correct

**Scope:**

4: The work is relevant to the Web and to the track, and is of broad interest to the community

---

### Official Review · Reviewer_GWpM · 2023-11-23

**Novelty:** 5
**Technical Quality:** 4

**Review:**

Dear authors,

thank you for you submission to WWW. I read the paper on access policies in a serverless function context with interest.

I would like to provide the following comments on the paper in its current form, and hope that they aid you in strengthening its presentation.

# Evaluation of GRASP

The paper states an evaluation set of 731 AWS Lambda applications supporting the first empirical study of serverless security policies. However, the paper also states that 90% (658) of those policies only included a single statement, with 92% of policies including 10 or fewer statements.

In addition, the paper does not provide information on the diversity of policies overall, i.e., whether the 658 policies including a single statement included--functionally--the same statement, and whether the remaining 10% of policies saw an overlap in the statements used, at least from a functional perspective.

As such, I am not convinced that the utilized evaluation methodology comprises a sufficiently diverse sample to support the strength of statements regarding the empirical contribution, or the evaluation of the proposed technique.

# Cloud system focus of related work

The related work discusses mostly items with a cloud focus. I would argue that various other fields also interacted with, what is essentially, ACLs and verifying their correctness/smallest fit for a given application or action, see for example, Nanevski, Aleksandar, Anindya Banerjee, and Deepak Garg. "Verification of information flow and access control policies with dependent types." 2011 IEEE Symposium on Security and Privacy. IEEE, 2011.

# Summary

Overall, i think that the presented work takes an interesting perspective on an emerging issue. However, it does not utilize a sufficiently large sample to evaluated the proposed implementation.

**Questions:**

- Can you provide overlap information between the classes (d1,d2,d3) and assessments (i.e., the examples before the list of contributions)?
- Can you provide information on the functional diversity of policies and statements?

**Ethics Review Description:**

-

**Reviewer Confidence:**

3: The reviewer is confident but not certain that the evaluation is correct

**Scope:**

2: The connection to the Web is incidental, e.g., use of Web data or API

---

### Official Review · Reviewer_Uvp9 · 2023-12-01

**Novelty:** 4
**Technical Quality:** 3

**Review:**

The work titled "Grasp: Hardening Serverless Applications through Graph Reachability Analysis of Security Policies" presents a new approach to enhancing the security of serverless computing applications.

### Strength
1. Comprehensiveness: The tool analyzes both application specifications and security policies, offering a holistic view of potential security risks.

2. Ease of Use and Extension: Grasp's design allows it to be easily extended to support additional frameworks beyond AWS Lambda, making it versatile.

3. Empirical Study: The empirical analysis of 731 AWS Lambda applications provides valuable insights into the common pitfalls in current serverless security policies.

### Weakness

1. It seems the author was in a hurry to submit their work, as there are some missing contents. The original text says, "This is a limitation of performing offline analysis on a large dataset and is discussed further in Section ??." However, there is no related discussion in the subsequent text.

2. Limitation in Offline Analysis: The tool's reliance on offline analysis limits its ability to handle dynamically resolved configuration values, which could affect the accuracy of its security policy analysis in certain scenarios.


In summary, the work on 'Grasp' is a high-quality and original contribution to the field of cloud computing security, specifically addressing the emerging challenges in serverless architectures. Its practical utility, combined with its methodological soundness, makes it a significant addition to current research. However, its certain limitations in handling dynamic configurations indicate areas for future improvement.

**Questions:**

1. Handling Dynamic Configuration Values:  How does Grasp address the challenge of dynamically resolved configuration values in serverless applications? Could you elaborate on the potential impact of this limitation on the accuracy of Grasp's policy analysis and any plans for future enhancements to mitigate this issue?

2. Scalability of Grasp: Could the authors provide more discussions about the scalability of Grasp in terms of handling larger and more complex serverless applications? Are there any performance benchmarks or limitations identified when analyzing extensive serverless architectures?

**Reviewer Confidence:**

3: The reviewer is confident but not certain that the evaluation is correct

**Scope:**

3: The work is somewhat relevant to the Web and to the track, and is of narrow interest to a sub-community

---

### Decision · Program_Chairs · 2024-01-22

**Decision:**

Accept

**Comment:**

## Summary
 GRASP introduces a graph-based framework for analyzing serverless access control policies in Amazon Lambda applications. The study evaluates 731 open-source applications, uncovering prevalent issues in serverless security policies, notably their over-permissiveness.

 ## Evaluation
 **Strengths:**
 1. **Comprehensive Analysis:** GRASP analyzes both application specifications and security policies, offering a holistic view of potential security risks.
 2. **Ease of Use and Extension:** Designed to be easily extended beyond AWS Lambda, GRASP has versatility.
 3. **Empirical Study:** Analysis of 731 AWS Lambda applications provides valuable insights into serverless security policy pitfalls.
 4. **Innovative Approach:** The use of graph-based analysis to assess serverless policies is a significant contribution to the field.

 **Weaknesses:**
 1. **Methodological Shortcomings:** The diversity of policies in the dataset is not clearly explained, affecting the strength of empirical contributions.
 2. **Dynamic Configuration Handling:** The tool's limitations in handling dynamically resolved configuration values could affect the accuracy of security policy analysis.
 3. **Lack of Real-World Validation:** The results from GRASP could be further validated with development teams or through user studies.
 4. **Limited Focus on Cloud Systems in Related Work:** The related work section predominantly focuses on cloud systems, overlooking relevant research in areas like ACLs and verifying their correctness.
 5. **Lack of Developer Feedback and Code Analysis:** There's no feedback from developers, and the analysis does not consider serverless code, only policies.

 **Suggestions:**
 1. **Expand on Methodology and Diversity:** Provide more details on the diversity of policies and statements in the dataset.
 2. **Dynamic Configurations Handling:** Explore methods to handle dynamic configurations in serverless applications.
 3. **Real-World Validation:** Engage with development teams for validation and conduct user studies to evaluate the impact of GRASP.
 4. **Broaden Related Work Discussion:** Include relevant research from areas intersecting with ACLs and policy correctness.
 5. **Incorporate Developer Feedback:** Seek feedback from developers and consider including serverless code in the analysis.

 **Overall Impression:**
 While GRASP marks a significant advancement in serverless computing security, particularly in policy analysis, enhancing its scope, addressing methodological gaps, and incorporating real-world validations will further strengthen its contributions to the field.

 ---